# Do the Benefits of School Closure Outweigh Its Costs?

**DOI:** 10.3390/ijerph19052500

**Published:** 2022-02-22

**Authors:** Elena Raffetti, Giuliano Di Baldassarre

**Affiliations:** 1Centre of Natural Hazards and Disaster Science, Uppsala University, 752 36 Uppsala, Sweden; giuliano.dibaldassarre@geo.uu.se; 2Department of Public Health and Primary Care, University of Cambridge, Cambridge CB1 8RN, UK; 3Department of Global Public Health, Karolinska Institutet, 113 64 Stockholm, Sweden

**Keywords:** COVID-19 pandemic, school closure, benefit-risk tradeoff, confirmation bias, precautionary principle

## Abstract

School closure has been a common response to COVID-19. Yet, its implementation has hardly ever been based on rigorous analysis of its costs and benefits. We aim to first illustrate the unintended consequences and side effects of school closure, and then discuss the policy and research implications. This commentary frames evidence from the most recent papers on the topic from a public-health epidemiology and disaster risk reduction perspective. In particular, we argue that the benefits of school closure in terms of reduced infection rates should be better compared with its costs in terms of both short- and long-term damage on the physical, mental, and social well-being of children and society at large.

## 1. Introduction

Childhood and adolescence are pivotal times for laying the foundations for a healthy life. Social interactions, intellectual stimulation, and the surrounding environment during school hours play a unique role in cognitive, physical, and psychosocial development [1,2]. Concurrently, education is not only recognized as a fundamental human right, but is also one of the key drivers for the achievement of the sustainable development goals (SDGs) towards a more equal and peaceful society [3]. Education has also latent functions including the facilitation of social mobility and gender equality, access to free meals for children from low economic status, and the prevention of child labor [4].

School closures have been a common response to the COVID-19 pandemic. They have affected up to 1.6 billion students worldwide, a total of 94% of the world’s student population from the beginning of the COVID-19 pandemic [5]. Up to 28 of November 2021, schools remained closed for 38,241,927 students (2.4%) and were only partially open for 700,559,055 students [5]. As a result, more than 1 billion children could not go to school for more than six months [5]. Despite schools reopening in many countries, parents remain hesitant to send their children back to school and prefer distance education [6,7].

This paper focuses on the benefits, side effects, and unintended consequences of school closure by discussing evidence from the most recent papers combining a public-health epidemiology and disaster risk reduction perspectives. This will be achieved by addressing the societal challenges that were identified in the UN Sustainable Development Goals (SDG). While a vast amount of the literature has focused on the side effects of school closure during the COVID-19 pandemic, there is little understanding of the impact of school closure. Specifically, this commentary aims to identify the research gaps, and proposes a research agenda for a better evaluation of the benefits, side effects, and unintended consequences of school closure.

## 2. The Benefits of School Closure

At the beginning of the COVID-19 pandemic, the choice to close down schools was primarily aimed at mitigating the spreading. Along with this, safeguarding children’s health was also a priority since SARS-CoV-2 was a new pathogen for human beings that stemmed from zoonotic transfer and there was uncertainty on its effects among children and adolescents [8,9]. The precautionary principle (“better safe than sorry”) [10] suggested that governments and public health agencies put in place all measures to protect future generations until the scientific community untangled the morbidity and mortality of COVID-19 infection among children as well as their role in the virus transmission. Models from previous epidemics of droplet infections, such as influenza, showed the role of schools as places that facilitate the spreading (Figure 1) and quantified the possible benefit that was obtained from school closure [11,12]. For example, previous studies concluded that school closure can be effective in case of severe influenza-like pandemics to reduce the burden on healthcare systems at the peak of the pandemic with an estimated reduction of 13–17% of the total number of cases and 39–45% peak attack rates [11].

Yet, the actual benefits of school closure during the COVID-19 pandemic are believed to be less than expected. First, children have been substantially less contagious than adults, as the vast majority of cases are mildly symptomatic and the mortality rates among children do not differ significantly from the ones of seasonal flu outbreaks [13,14,15]. Second, already in August 2020, evidence supported a marginal role of children as drivers of the first wave of COVID-19 infections. A recent simulation on reopening primary schools in the UK, a country with 18% individuals over 65 years of age, shows a low risk of outbreaks also considering data from the second wave [16]. Along with this, school reopening did not play a causal role in the increased number of new cases and hospitalization in Italy during the period of September–October 2020 [17], and it was not the main driver of the increased spreading of SARS-CoV-2 lineage B.1.1.7 in England (COVID-19 UK variant) at the beginning of 2021 [18].

In the dynamic transmission model for influenza (flu) outbreaks that was suggested by Cauchemez [11], schools are often seen as drivers of outbreaks (Figure 1A). Yet, the analysis of COVID-19 outbreaks shows that secondary cases in schools are the most dominant feature (Figure 1B). This is due to differences between the flu and SARS-CoV-2 virus. The flu virus is characterized by eight single-stranded RNA that facilitates mutations along with antigenic shift and drift of surface proteins (i.e., hemagglutinin H and neuraminidase, N) [19]. This makes the elderly more protected than the younger generations during outbreaks since they have been exposed to a higher number of flu virus genotypes in a lifetime. At odds, SARS-CoV-2 is a new pathogen for human beings and causes milder symptoms in the young generation, mainly due to a lower response of the immune systems [15]. While testing random samples of the general population and the subsequent contact tracing may help to uncover the context specific dynamic transmission models, uneven testing between schools and other social settings may bias the results and reflect a higher incidence of COVID cases in the school setting.

## 3. The Costs of School Closure

Closing schools for prolonged periods (e.g., several months) is associated with a range of short- and long-term risks for the health of children and society at large [20]. School closures during past outbreaks and summer vacations have been associated with a setback in physical, social, and mental well-being mainly in children with low socioeconomic status [21,22]. Being at home increases the risk of abuse or neglect along with losing the opportunity of the only daily meal [20,23,24]. In the European Union, more than two million children face food insecurity [25]. Moreover, the effects on mental health and well-being are devastating. For example, a nationally representative study in Germany showed an increase of depressive and anxiety symptoms (from 10% to 18% prevalence) among children and adolescents during the ongoing pandemic, particularly among those with low socioeconomic status, limited living space, and migration background [26]. Beyond the direct effect on children’s well-being and development, lacking opportunities to socialize and learn increases the long-term risk of dropouts, substance abuse, psychiatric problems, and chronic diseases among children leading to a rise of socioeconomic and gender inequalities [27,28,29]. Despite many countries having made efforts to move to distance learning, distance learning cannot be a satisfactory solution as face–face lessons are irreplaceable in terms of social interaction and learning. Children from low economic status and living in rural areas may have no (or limited) internet access, technological, and parental support for educational purposes. Figure 2 summarizes the side effects of prolonged school closure for children and society in the short- and long-term.

Moreover, school closure has unintended consequences on the spreading of COVID-19. The balancing (B) loop of Figure 3 shows the intended benefits of school closure, which consists of reducing the spreading of the pandemic and thus COVID-19 cases and deaths. Yet, there are a number of unintended consequences that can offset the initial benefits of school closure and paradoxically increase COVID-19 deaths. To illustrate, we provide three examples in Figure 3, see reinforcing (R) loops. While school closure may marginally contribute to slowing the spread of COVID-19, child-care obligations can lead to healthcare workers’ stress and absence, home-based care that often relies on the elderly, and in turn, to higher mortality. In the long term, school closure may also increase gender and socioeconomic inequalities and thus affect the society at large. Specifically, school closure requires caregivers to stay home. Parents, most often mothers or grand-parents, must take care of the children. While the former option led to an exacerbation of gender disparities [30], while the latter has paradoxically exposed the most vulnerable people (i.e., the elderly) to COVID-19 and arguably raised COVID-19 mortality. Second, school closure leads to a reduction of the number of essential workers, as healthcare staff, lowering the resilience of healthcare systems, and household income. For example, a study in the US estimated 2.0–2.35% increased mortality rate when the healthcare workforce declined by 15% due to school closure [31]. Third, the increased inequalities that stem from school closure may raise the proportion of vulnerable people in the long-term, which again increases COVID-19 mortality (Figure 3, dashed R loop). Wildman (2021), for instance, found that in OECD countries a 1% increase in income inequality (Gini coefficient) is associated with an approximately 5% increase in COVID-19 deaths [32].

In summary, the choice of closing schools can also be framed within the UN Sustainable Development Goals (SDGs). While marginally reducing the spreading, school closure has substantial impacts on the quality of and access to education (SDG 4, quality education), increase social vulnerability (SDG1 no poverty, SDG 2 no hunger, SDG 5 gender equality), and impact community health and wellbeing (SD3 good health and wellbeing).

## 4. The Role of Scientists

Although the main beneficiaries of education are children, the responsibility to protect children’s rights is societal and lies in everyone’s interest. As such, scientists should inform the decision-making process by developing and integrating scientific knowledge about the benefits of school closure in terms of reduced transmission rates as well as their costs, i.e., the short- and long-term damage on children and society at large.

Yet, the aforementioned findings on the limited benefits of school closure on the spreading and children’s well-being have not been effectively disseminated in traditional- and social media platforms. For example, among the 100 most-discussed scientific articles in 2020 (Altmetric Top 100), 25 articles focused on the COVID-19 pandemic (7 in the first 10 positions) [33]. Among them, only two articles discussed how children and schools are impacting the spread [14,34]. These figures do not seem to match the (urgent) relevance of this issue. Lastly, it is worth mentioning that three articles examined the role of non-pharmacological interventions on COVID-19 spreading, grouping school closure together with other measures [35,36,37]. Strikingly, none of these papers discussed the disruptive effects of school closure on children’s mental and physical health.

To best inform the policy-making process, more research is needed on the side effects of school closure on other aspects of public health (Figure 2B), as well as on potential unintended consequences on the same COVID-19 mortality (Figure 3). More knowledge on short- and long-term effects will enable more rigorous analysis of the costs and benefits of school closure as a response to the ongoing pandemic.

## 5. Context-Specific Policy

The ratio between the benefits and costs of school closure unavoidably depends on the local context. The preparedness of healthcare systems, prompt response to the spreading, and characteristics of the population have a determinant role on the impact of pandemics. The initial COVID-19 outbreaks in the Wuhan province (China) and Lombardy region (Italy) highlighted that public healthcare systems could not cope with a high number of severe cases in a short period [38]. A prompt response to the spreading adapted to the characteristics of the population, such as the proportion of elderly, comorbidity, and intergenerational households along with the population density, is of importance to mitigate the COVID-19 pandemic. For example, not having resilient healthcare systems and a high prevalence of communicable comorbidities (HIV and tuberculosis) led many African countries to close schools as a common prompt response to epidemic outbreaks. For example, during the Western Africa Ebola outbreak, schools were closed for more than seven months and were often used as treatment centers [22].

However, the benefit of closing schools during the current pandemic should be cautiously examined. A younger population in the majority of African countries compared to Europe led to a marginal excess mortality due to the COVID-19 pandemic [39], while the consequences of school closure are crushing on children’s health, food security, social and gender equality, and economic growth. Another prominent negative example is the school closure strategy in Latin America and the Caribbean. While it is the region with the longest period of school closure [5], several countries in the region have the highest case fatality rate globally [40].

Besides the age distribution, the ratio between the costs and benefits of school closure depends on the prevalence of intergenerational households. In Europe, for example, these are more common in the Mediterranean than in the Nordic countries. For example, in Sweden, regardless of schools that remained open for children up to 13–14 years of age throughout the pandemic, 80% of the first confirmed cases in households were an adult, while children were only secondary cases, primarily infected at home [14].

Finally, policies could be adapted to the year groups. When considering the fundamental role of socialization on cognitive development and learning during childhood and the increased risk of transmission with age, governments can implement distance learning for high schools and university education if needed, while elementary schools may be maintained open.

## 6. Conclusions

Prolonged school closures around the world have not been based on compelling analyses of their costs and benefits. Instead, decisions have been driven by common tendencies and confirmation bias in crisis management. Short-term solutions that are based on linear thinking still dominate the decision-making process [41]. As a result, most authorities tend to prioritize measures that are likely to work in the short-term (e.g., the B loop in Figure 3), but can backfire in the long-term (e.g., the three R loops in Figure 3). These shortsighted measures, which are also known in the literature as “fix that fails” [42], are yet most popular and thus the first choice for most politicians.

Public concerns and decision-making processes are more easily driven by countable threats, such as the COVID-19 pandemic where the daily number of COVID cases and deaths is quantified and communicated on a regular basis (daily). Conversely, similar responses do not stem from aspects that are more difficult to quantify, such as the risk of mental health problems or neglect among children and the negative effects on society at large.

We argue that a wise application of the aforementioned precautionary principle should not focus on a single threat. As education plays a fundamental societal role, the principle also advocates that schools should remain open as much as possible to guarantee students’ access to education and prevent devastating effects (Figure 2B) on children and society at large. This is challenging during a major crisis as people tend to internalize the experience and assess the risk of epidemics as more likely and impactful [43]. Thus, scientists should help policy-makers in providing a holistic understanding of the effects of public health interventions. As argued in this commentary, school closure as a response to COVID-19 has been often evaluated in terms of its positive effects on the spreading of the pandemic. Instead, we posit that these non-pharmaceutical interventions should be deemed as drugs in clinical medicine with an explicit consideration of both the benefits and side effects. While short- and long-term risks of school closure have been discussed in the literature, the relevance of side effects (Figure 2B) and unintended consequences (Figure 3) across different contexts is still unclear. This requires advances on several fronts. First, national monitoring systems that actively track children’s wellbeing should be improved. Register-based data are needed to better quantify how many children from low socio-economic status do not have access to free meals or face barriers to access healthcare services, and how many children live in areas with a high criminality rate, or in families that are followed by social services for domestic violence. Moreover, we posit that national surveys should be regularly (e.g., yearly) implemented in school systems to monitor the trends of children and adolescents’ wellbeing. This will allow to quantify several domains of wellbeing and overcome the limitations of the actual monitoring systems. Second, we call for multidisciplinary research to better evaluate the costs and benefits of school closure on the basis of population characteristics, type of national policy response, and preparedness of the healthcare system. There are ongoing discussions in many disciplines about the potential benefits of integrating multiple sources of data (e.g., interviews, surveys, biological measures, register-based data) and methodological approaches (e.g., longitudinal and quasi-experimental study) to increase the reliability of scientific findings and address public concerns. These evaluations should not only consider monetizing tangible losses (when possible), but also exposing the intangible costs that cannot be fully expressed in monetary terms, such as increasing domestic violence or gender inequality. Third, decision support systems should explicitly include the short- and long-term effects of school closure in order to better inform the policy-makers and improve the planning of emergency responses. The quantification of the unintended consequences and side effects that are depicted in Figure 2, for example, can enable the inclusion of these feedback mechanisms in mathematical (epidemiological) models and thus increase their usefulness. Finally, school closure is often considered as an option by societies that are facing social-environmental extreme events. The implemented efforts and lessons that are learned during the current pandemic, such as moving to distance learning, should be considered when defining preparedness plans. More specifically, improvements in internet access and technological support in rural areas, mentoring programs for children from low socio-economic families, and the evaluation of implemented context-specific interventions to mitigate the effect of school closure should be included in governmental agendas. Analyzing the benefits and costs of school closure, along with their context-specific unintended consequences and side effects, is key to best inform policy-makers and help them account for the complexity of health, as “a state of complete physical, mental, and social well-being” (World Health Organization), when responding to public health crises.

## Figures and Tables

**Figure 1 ijerph-19-02500-f001:**
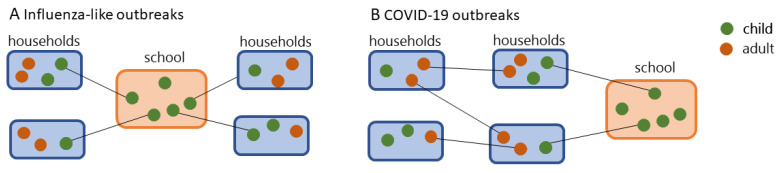
Transmission-dynamic scenario for (**A**) influenza (flu)-like outbreaks as described by Cauchemez [11] and (**B**) COVID-19 outbreaks.

**Figure 2 ijerph-19-02500-f002:**
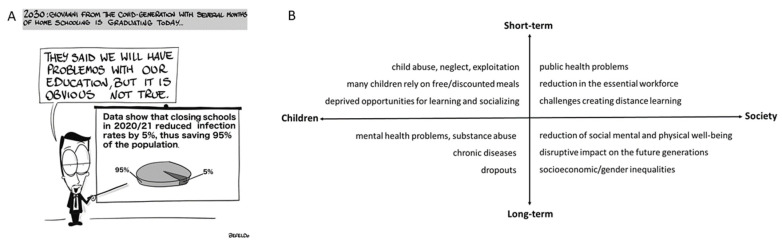
(**A**) School closure and the right to education. Deprived opportunities for learning are only one of the negative effects of school closure on children. (**B**) Side effects of school closure for children and society in the short- and long-term.

**Figure 3 ijerph-19-02500-f003:**
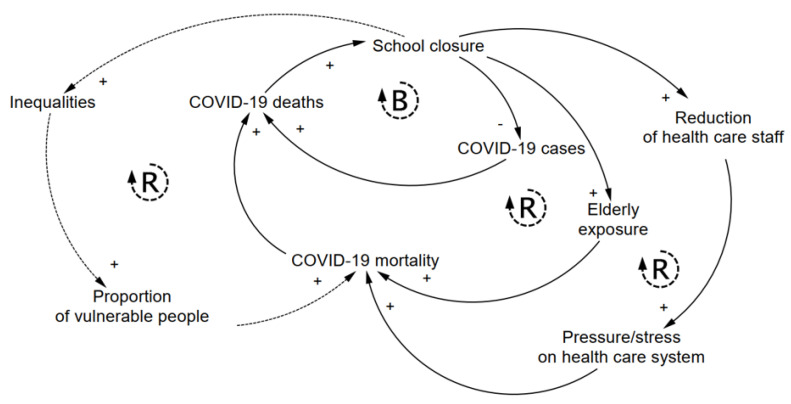
Unintended consequences of school closure. The balancing loop (B) schematizes the intended benefit of school closure to reduce the infection rates and thus COVID-19 cases and death. The three reinforcing loops (R) schematize examples of the unintended consequences (elderly exposure, reduction of healthcare staff, and I ncreasing inequalities) that can offset the intended benefit and paradoxically increase COVID-19 mortality and deaths. System dynamics notation is used to indicated positive (+) and negative (-) feedbacks. Dashed lines are used for long-term effects (e.g., increasing inequality).

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
