# Peer review of "Do the Benefits of School Closure Outweigh Its Costs?"

_ijerph, 2022, doi:10.3390/ijerph19052500_

Round 1

Reviewer 1 Report

I suggest revision for the paper in order to reach out scientific paper standards

  1. Abstract should be enriched (aim, method, results)
  2. Methodology part of the paper needs to be enriched
  3. Research questions and the need for the study need to be added
  4. Conclusion needs argumentation by research questions.

Reviewer 2 Report

The main objective of the article is illustrate the unintended consequences and side 14 effects of school closure as a way to cope with COVID-19, and then discuss policy and research 15 implications.

The paper is well written and is composed and structured by all the main parts of a scientific article need.  

If possible, improve the presentation of the methodology, either in the summary and introduction, or in the addition of a section on the methodology. 

If you agree, add some more information above the figure 2 to explain better the unintended consequences of school closure.

Reviewer 3 Report

As a commentary, I think this article is fine. However, I encourage the authors to strength the introduction and overall discussion of the article. First, I recommend the author highlight the existing debate about school closure in the literature and/or among the public and suggest how the article may contribution to the debate. Second, as an academic commentary, I expect everyone can gain theoretical insight from the article. But, I only find the authors only summarizes and describes existing and previous observations and simply list the pros and cons of school closure. I think this writing practice is fine if it is a editorial published by newspaper or blog, but not academic journal article. Therefore, I encourage the authors to discuss the issue based on theoretical framework or perspective in order to deepen our understanding of the effectiveness of school closure.

Round 2

Reviewer 3 Report

I think the manuscript is improved significantly. Congratuations!